# Unveiling the Potential Role of Nanozymes in Combating the COVID-19 Outbreak

**DOI:** 10.3390/nano11051328

**Published:** 2021-05-18

**Authors:** Jafar Ali, Saira Naveed Elahi, Asghar Ali, Hassan Waseem, Rameesha Abid, Mohamed M. Mohamed

**Affiliations:** 1Civil and Environmental Engineering Department, United Arab Emirates University, Al Ain 15551, United Arab Emirates; jafarali@uaeu.ac.ae; 2National Water and Energy Center, United Arab Emirates University, Al Ain 15551, United Arab Emirates; 3Department of Biochemistry and Molecular Biology, University of Sialkot, Sialkot 51310, Pakistan; sairanaveed292@gmail.com (S.N.E.); hasan_wasim@hotmail.com (H.W.); 4FMH College of Medicine & Dentistry, Lahore, Punjab 54000, Pakistan; drasgharali1228@gmail.com; 5Department of Biotechnology, University of Sialkot, Sialkot 51310, Pakistan; Ramesha.255@gmail.com

**Keywords:** coronavirus disease 2019 (COVID-19), SARS-CoV-2, nanozyme, diagnostics, vaccines

## Abstract

The current coronavirus disease 2019 (COVID-19) outbreak is considered as one of the biggest public health challenges and medical emergencies of the century. A global health emergency demands an urgent development of rapid diagnostic tools and advanced therapeutics for the mitigation of COVID-19. To cope with the current crisis, nanotechnology offers a number of approaches based on abundance and versatile functioning. Despite major developments in early diagnostics and control of severe acute respiratory syndrome coronavirus 2 (SARS-CoV-2), there is still a need to find effective nanomaterials with low cost, high stability and easy use. Nanozymes are nanomaterials with innate enzyme-like characteristics and exhibit great potential for various biomedical applications such as disease diagnosis and anti-viral agents. Overall the potential and contribution of nanozymes in the fight against SARS-CoV-2 infection i.e., rapid detection, inhibition of the virus at various stages, and effective vaccine development strategies, is not fully explored. This paper discusses the utility and potential of nanozymes from the perspective of COVID-19. Moreover, future research directions and potential applications of nanozymes are highlighted to overcome the challenges related to early diagnosis and therapeutics development for the SARS-CoV-2. We anticipate the current perspective will play an effective role in the existing response to the COVID-19 crisis.

## 1. Introduction

In the 20th century, millions of deaths occurred due to three influenza pandemics [1]. The first deadly pandemic named ‘Spanish flu’ occurred in 1918, which was caused by H1N1 influenza A strain. The estimated number of deaths from this virus were more than 40 million [2]. In 1957 another global pandemic called ‘Asian Flu’ [3,4] first originated in the Yunan province of China [4]. This virus was a mutated form of avian (H2N2) and already present human influenza viruses strains [3]. In mid-1968, the Hong Kong flu pandemic occurred, which was caused by the H3N2 influenza strain and afterward spread all over the globe [1,4]. The death toll of this pandemic was approximately 700,000 [4]. Recently, in late December 2019, a novel coronavirus (CoV) outbreak was first reported in Wuhan, China. The virus first disseminated in different parts of the world and eventually gained the status of a pandemic. The World Health Organization (WHO) announced this outbreak as a worldwide emergency on 30 January 2020 [5]. Etiologically, this illness has been caused by severe acute respiratory syndrome coronavirus 2 (SARS-CoV-2) [6,7]. Up to April 2021 more than 2.91 million deaths have been reported.

SARS-CoV-2 is an enveloped, positive sense, single-stranded RNA beta-Covid encoding 3 non-structural (3-chymotrypsin-like protease, papain-like protease, helicase, and RNA-dependent RNA polymerase), structural (spike glycoprotein) and accessory proteins [8]. SARS-CoV-2 is highly infectious and human-to-human transmission has been frequently reported. The world economy is facing a long lasting downturn due to mandatory quarantine and lockdowns. The implications of the pandemic are worsening day by day which poses serious challenges for health services. One serious concern of healthcare professionals is to develop, disseminate and deploy safe and effective vaccines against COVID-19 [9]. Until now, few SARS-CoV-2 vaccines have been approved, with more expected to be licensed in Yet having licensed vaccines, efficacy, large scale production, affordable price and rapid availability to local communities remain big challenges in controlling the SARS-CoV-2 pandemic. Reinfections and uncertainties in preliminary data for some vaccines indicate that the immunization process needs further investigation.

Meanwhile, reports about the emergence of new variants of coronavirus are continuously rising [10]. Early diagnosis and quarantine to cut off the source of infection are the most effective control strategies for disease outbreaks, including the COVID-19 pandemic [11,12]. Diagnosis of SARS-CoV-2 might be influenced by epidemiological history, clinical features, imageology and pathogenic index. Development of effective antiviral agents is hindered by the ability of SARS-CoV-2 to grow in the host cells without keeping its genome. Thus, there is an urgent need to expand testing capacities, deploy effective therapeutics, and develop safe vaccines that provide long lasting immunity [13]. Reverse transcription polymerase chain reaction (RT-PCR) is the most common molecular method used for early detection of SARS-CoV-Some other nucleic acid-based techniques, such as CRISPR (clustered regularly interspaced short palindromic repeats), microarrays, high-throughput qPCR (HT-qPCR) and loop-mediated isothermal amplification (LAMP) are also favorable options for detecting SARS-CoV-2 in clinical as well as in environmental samples [14,15,16]. Despite reliable results, genome extraction, amplification and data analysis require sophisticated biosafety labs, skilled personnel that make nucleic acid testing costly and unsuitable for under-developed countries [17]. Alternatively, antibody testing is chosen for detecting IgM or/and IgG antibodies produced after exposure to SARS-CoV-2 [18]. Typically, production of antibodies occurs 10–15 days post infection, so early screening and diagnosis of SARS-CoV-2 could not be possible with this technique [19]. The WHO has recommended the use of rapid antigen diagnostic tests that meet at least 80% sensitivity and 97% specificity for the active SARS-CoV-2 infections [20]. A detailed overview of various currently used diagnostic techniques, with detection limits, specificity, processing time is presented in Table 1. It could be inferred that extensive efforts are required to improve the early diagnosis of SARS-CoV-2, which will improve the therapeutic decision-making and will further decrease the intensity of illness and duration of hospital stay.

Various biosensing techniques have been developed for rapid, reliable, and sensitive detection of biomolecules (biomarker) for gauging virulence, pathogenicity, and microbial load [21,22]. Biosensing methods use natural enzymes such as horseradish peroxidases to catalyze different colorimetric reactions in the presence of substrates. Regardless of their novel catalytic effectiveness, natural enzymes have some limits for industrial application, such as low stability in harsh environmental conditions, and relatively high costs for preparation, purification, and storage [23]. Therefore, over the past few decades, researchers have made an intense effort to develop artificial enzymes for a wide range of applications [24]. Recently, nanomaterial-based enzyme mimetics (nanozymes) have revolutionized the fields of diagnosis and therapeutics [24].

Nanozymes have been frequently employed in various biomedical applications such as disease diagnosis, cancer therapy and anti-viral agents. Therefore, inspired by the unique characteristics of nanozymes, it is assumed they have the potential to overcome the challenges related to the early diagnosis and therapeutic developments for SARS-CoV-2 infections. According to the author’s knowledge, almost no efforts have been devoted to reviewing the vast potential of nanozymes to combat the COVID-19 infection. Thus, in this perspective, we present a comprehensive study of recent updates on nanozymes and their possible applications for detection and treatment for the SARS-CoV-2. We anticipate the current article will pave the way toward the development of rapid and sensitive diagnostics. Moreover, our findings will play a very effective role in the welfare of humans and the medical community in the COVID-19 crisis.

## 2. Overview of Previously Used Nanozymes in Various Antiviral Therapies

Nanomaterials with enhanced biocatalytic properties (nanozymes) have effectively contributed to overcome the limitations of natural enzymes. Owing to their unique characteristics, nanozymes have attracted enormous attention from the scientific community [66,67]. Nanozymes display several remarkable advantages such as high stability, reusability, low cost, easy fabrication, and versatility. Various catalytic properties i.e., catalase, oxidase, peroxidase, and superoxide dismutase have been widely explored for triggering cascade biochemical reactions for biomedical applications (Figure 1). Moreover, tunable and intrinsic physicochemical properties make nanozymes a promising candidate for antibacterial and antiviral agents. Nanozymes can improve the sensitivity and quantitative potential of biomedical devices used for viral infections. The use of nanozyme during different outbreaks caused by Nipah virus, Ebola virus, Dengue virus, Coronavirus, Chikungunya virus, and Influenza virus is well documented [13]. Biomedical applications of nanozymes (nanomaterials) range from imaging, detection of antigen, bioassays and inactivation of viruses/bacteria [68,69]. We have summarized the previously reported, biomedical and therapeutic applications of nanozymes for various viral infections (Table 2).

## 3. Contribution of Nanozymes for Combating Severe Acute Respiratory Syndrome Coronavirus 2 (SARS-CoV-2)

Nanozymes have emerged as a suitable alternative for both therapy and diagnosis of SARS-CoV-2. Extraordinary stability and high catalytic efficiency at nanoscale qualify the nanozyme-mediated strategies as a powerful approach for tackling SARS-CoV-2. Although various nanoparticle (NP)-based approaches have been decisive for the development of disinfectants, their role in new testing and diagnostic kits for SARS-CoV-2 infection has also been realized [13,78]. In fact, nanozymes can be used for potential new therapeutics and improved anti-viral activity for SARS-CoV-2 [79]. High-sensitivity and point-of-care testing (POCT) can be obtained by nanozymes for SARS-CoV-2 antigen detection [71] and nano-carriers can build risk-free and effective immunization strategies for SARS-CoV-2 vaccine candidates, for example, nucleic acids and protein construct [80]. Thus, nanozymes can make an effective contribution in the fight against the COVID-19 pandemic in all four key areas of action, that is, point-of-care detection, surveillance and monitoring, therapeutics, and vaccines. In the following part of the perspective, we have described the different possible contributions of nanozymes for the fight against SARS-CoV-2 in different ways (Figure 1).

### 3.1. Nanozymes in the Detection of SARS-CoV-2

Different molecular and serological techniques are being used to detect the SARS-CoV-2 infections and resulting antibodies (Table 1). Nanomaterials are an essential component in some of these technologies because they play an important role in the detection/transduction of biochemical interactions [79]. Previously, nanozymes were successfully used in a colorimetric strip test for the effective detection of Ebola virus [72]. Similarly, a nanozyme-based chemiluminescence paper test for rapid, highly sensitive, and portable detection of SARS-CoV-2 spike antigen is reported. In this study, Co-Fe@hemin nanozyme was used as a suitable alternative to horseradish peroxidase (HRP) for providing the point-of-care testing (POCT) for SARS-CoV-2. Briefly, traditional enzymatic chemiluminescence analysis was combined with a lateral flow assay to achieve the detection limit of 0.1 ng/mL in 16 min. Despite high specificity and sensitivity for recombinant antigen and pseudo-virus, a strong validation of nanozyme-strips against clinical samples is required [71]. A parallel comparison of this strip test with other commercial kits is still needed to commercialize the nanozyme-mediated detection strategies. Futuristic investigations must opt quantitative approach and should also develop a portable handheld device equipped with signal acquisition and data analysis features.

An ultrasensitive colorimetric assay called magnetic nanozyme-linked immunosorbent assay (MagLISA) was used for the detection of Influenza A (H1N1) virus. Primarily, silica-shelled magnetic nanobeads (MagNBs) and gold nanoparticles were combined to detect the 10^−15^ g/mL of antigen. Signal amplification and peroxidase-like activity was performed by gold nanozymes (AuNZs). Inspired by the above studies, the sensitivity of serological tests (enzyme-linked immunosorbent assay (ELISA)) for SARS-CoV-2 can be improved by nanozymes [73]. Since the vaccination campaign has started in many countries, antibody testing through ELISA has been ruled out due to low sensitivity and incapability to detect the antigenic components of SARS-CoV-2. It is anticipated that indirect/sandwich ELISA for spike proteins SARS-CoV-2 will certainly improve the precision diagnoses as well as characterization of the spread and prevalence of the COVID-19 disease (Figure 2). Thus, nanozyme-based conjugates for indirect ELISA can lower the overall cost and burden from molecular techniques used in the detection of SARS-CoV-2.

Nanozymes have been used as a model for POCT strategies because of their efficient catalytic potential to use in clinical diagnosis and detection of SARS-CoV-2. In this regard, nanozyme-mediated paper-based lateral flow assays (LFAs) can be considered as an effective approach [81]. Previous studies have shown that the plasmonic properties of AuNPs enabled the direct utilization of nanomaterials to obtain colorimetric readouts of LFAs [82]. Nanozymes are inserted to enhance the performance and sensitivity of paper-based biosensors. An additional chromogenic substrate made the nanozyme-based LFAs different from conventional AuNP-based LFAs. Therefore, in the presence of the target, the reaction between the nanozyme complex and chromogenic substrate eventually generates a strong colorimetric signal from the respective color-producing reaction compared with that of conventional LFAs [70]. Asymptomatic patients and cross reactivity of SARS-CoV-2 antibodies with antibodies of other coronaviruses may result in false measurements. Therefore, accurate detection strategies are needed for the diagnosis of a highly pathogenic strain of COVID-19. A recent study reported the ultrasensitive detection of SARS-CoV-2 related antigens based on an aptamer-assisted proximity ligation assay. In this method, the ligation DNA region was brought into the vicinity to initiate the ligation dependent (qPCR) amplification by the two aptamer probes binding on the same (protein) sensor. Enhanced specificity and rapid detection mainly depend on the interaction between spike S1 (SARS-CoV-2) and its receptor ACE2 (human cells) [83]. It is expected that nanozyme mediated/engineered aptamers could be useful in the development of effective and affordable therapeutics and prophylactic vaccines for SARS-CoV-2 and other infectious pathogens [84].

### 3.2. Nanozymes for Inactivation/Inhibition of SARS-CoV-2

Inactivation and inhibition of SARS-CoV-2 are necessary to combat the worst pandemic of COVID-19. Much is still unknown about SARS-CoV-2, but the following facts have been established, i.e., it is an enveloped ssRNA virus, with four integral proteins: (i) the S protein (spike glycoprotein) involved in the attachment of the virus to host cells; (ii) the M protein that maintains the membrane integrity of the viral particle; (iii) the E protein (envelope) is the smallest protein and plays a structural role and helps in assembly and budding; and (iv) the N protein (nucleocapsid) predominantly binds to the SARS-CoV-2 RNA and supports nucleocapsid formation [80]. Coronavirus infections start with the binding of virions to cellular receptors. The major steps involved in the SARS-CoV-2 replication cycle are (1) attachment and entry of SARS-CoV-2 to the host cell, (2) release of viral genome in host cells, (3) translation of viral polymerase proteins, (4) RNA replication, (5) sub-genomic transcription, (6) viral structural protein translation, (7) viral structural proteins attach with nucleocapsid, (8) development of mature virion, and (9) release of mature virion through exocytosis. The newly produced virions further infect new individuals and the same cycle continues [85]. Recently, numerous research groups have suggested the use of diverse nanomaterials to control the spread of SARS-CoV-2. Nanozyme-mediated therapeutic drugs can inhibit the effects of viral infection in several ways, including by blocking receptor binding and cell entry, inhibiting the viral replication/proliferation, and directly inactivating the virus. Here are some proposed ways that can inhibit SARS-CoV-2.

#### 3.2.1. Inhibition of Viral Entry

During the attachment phase of SARS-CoV-2, the S1 protein specifically binds to the human-converting enzyme 2 (ACE2) receptor present on the surface of human cells [86]. Entry of the virus in the cell/membrane is facilitated through endocytosis. Hence, inhibiting the virus endocytosis by nanozymes will be a potential target to inhibit SARS-CoV-2. Nanozymes could directly interfere with membrane fusion of viruses. Previously, iron oxide nanozymes (IONzymes) were used for catalytic inactivation of enveloped influenza viruses. Inactivation mechanism involved the lipid peroxidation of the viral lipid envelope and disruption of the nearest proteins hence preventing the transmission and infection of the virus. Similarly, the inactivation spectrum of IONzymes could be extended for SARS-CoV-2 which is also an enveloped virus. Another study reported the graphene QDs could interfere the cell binding of HIV (human immunodeficiency virus), feline coronavirus (FCoV) and other enveloped viruses entry was blocked by AgNPs combined with graphene oxide (GO) sheets [87]. Moreover, surface-functionalized metal nanomaterials/nanozymes i.e., AgNPs and AuNPs were used to block the entry of HIV and herpes simplex virus (HSV) [88]. Previous antiviral applications of metal NPs and structural similarities of enveloped HIV/HSV with COVID-19 mean that it could be inferred that gold-silver core nanozymes can provide extended antiviral activities against the SARS-CoV-2. Nanozymes may attach to the S protein active site and ultimately blocks the binding between S-protein of virus and ACE2 receptor of a host cell, hence preventing the spread of infectious SARS-CoV-2. A possible mechanism of action for nanozyme is shown in Figure 3. Conclusively, development of an effective, broad-spectrum and multimodal nanozymes with antiviral activity will provide an upper hand against SARS-CoV-2 and other emerging viruses.

#### 3.2.2. Blocking the Synthesis of Viral RNA and Viral Assembly

After the entry of the virus into the host cell, inhibition of replication/ infectivity through nanomaterials is considered an effective therapeutic control. The mechanism behind disrupting the virus replication cycle is to stop the activity of RNA polymerase and cease the transcription (Figure 3). These treatments keep the virus number limited so that it can easily be controlled by the immune system. Several nanoparticles have been investigated for the inhibition of enveloped coronaviruses. For example, zinc salt was used to inhibit hepatitis E virus replication via blocking the transcription of virus. Previously, ZnO nanomaterials have been extensively studied for antimicrobial applications. Therefore, Zinc based nanozymes could be synthesized to release Zn^2+,^ which may bind to RNA polymerases and result in the inhibition of RNA transcription of SARS-CoV-2. The infectivity of several coronaviruses has been diminished using silver nanoparticles (AgNPs) and silver nanowires (AgNWs) with the concentration below the toxic level. The proliferation inhibition was achieved by suppression of RNA synthesis (Figure 3) [89]. Another study suggested that glutathione coated AgS_2_ nanoclusters were promising antiviral therapeutic agent for model corona virus called porcine epidemic diarrhea virus (PEDV). Investigation showed that PEDV infection was suppressed about 3 times after 12 h post infection. The antiviral mechanism of nanoclusters explained that viral negative strand RNA synthesis and budding was halted [90]. Moreover, interferon and cytokine production were further stimulated to completely stop the infection of coronavirus. The above findings are indicating the involvement of more than one pathway in viral inactivation, so these might serve as guides for research and development on SARS-CoV-2. Thus, a silver-based nanozyme could deliver enhanced antiviral efficacy by simultaneously inhibiting the entry of new and blocking the replication of already infected SARS-CoV-2.

A multisite viral inhibitory mechanism of glycyrrhizic-acid-based carbon dots (Gly-CDs) was recently reported for the porcine reproductive and respiratory syndrome virus (PRRSV) and pseudorabies virus (PRV). Extraordinary antiviral activity was achieved by inhibition of viral invasion and replication, stimulation of IFN production in cells, and inhibition of viral-infection-induced ROS production. In vivo application of Gly-CDs for inhibiting SARS-CoV-2 might be challenging due to biocompatibility issues. Carbon-based nanozymes can improve the virus and nanomaterial interactions [91]. Recently, a combination of hypericin-graphene oxide (GO) was used against the reovirus [92]. Interestingly, hypericin is also one of the computationally identified drug for the SARS-CoV-2. In summary, a carbon nanozyme such as GO has the immense potential to contribute to fight against SARS-CoV-2.

Instead of inhibiting the cell−virus attachment, or genetic material replication, another strategy to halt viral infection is inactivation/destruction of the virus itself. The majority of nanozymes may have a high affinity towards virus attachment due to nanoscale dimensions. They can interact and directly inactivate/destroy the viral structures without harming the host cells [89,93]. Gao and associates showed the catalytic inactivation of iron oxide (Fe_3_O_4_) NPs targeting the viral envelopes on 12 diverse subtypes (H1−H12) of IAVs. The ferromagnetic Fe_3_O_4_ NPs with an average of 200 nm were named iron oxide nanozymes (IONzymes) because of their enzyme-like properties, catalyzing peroxidase, and catalase responses. Subsequently, IONzymes could firmly instigate lipid peroxidation in the viral envelope and break the integrity of viral surface proteins, including HA (haemaglutinin), neuraminidase, and matrix protein I, leading to inactivation of the virus. Furthermore, the scientists stacked the IONzymes on facemasks and observed the great protection against different strains of IAVs, including H1N1, H5N1, and H7N. The high biocompatibility and easy synthesis of the IONzyme nanomaterial make these attractive for successful and safe beginning phase antiviral therapeutics. Accordingly, the IONzyme present a possibly ground-breaking approach in fight against SARS-CoV-2 [77,89].

#### 3.2.3. Vaccine Approach

Effective vaccines are vital to attain the desired “herd immunity” and to completely control the pandemic. Innate and adaptive immune systems work in coordination to eliminate the SARS-CoV-2 from infected cells. Overall, the antiviral mechanism of immune system involves the production of antibodies, IFN-γ, interleukins (ILs), and tumor necrosis factor-α (TNFα), two important types of cytokines for antiviral immune responses [89]. It is expected that an effective vaccine against SARS-CoV-2 should be immunogenic and safe. Long-term immune response and balanced activation of T and B cells remain the prerequisite of reliable vaccines [94]. Nanotechnology-based vaccines are preferred due to their enhanced potential to be properly exposed to the immune system, protection against nucleases and proteases along adjuvant activity. Among the currently approved vaccines for SARS-CoV-2, the BNT162b2 (Pfizer/BioNTech) is encapsulated in lipid nanoparticles [95]. This nano encapsulation reduces the potential risk of integration of exogenous DNA into the host genome. Apart from structural entities nanozymes/nanomaterials can act as delivery vehicles, trigger the immune system and improve the thermostability for distribution and availability [96]. Previously, iron oxide nanozymes (IONzyme) were successfully used for enhanced catalytic mucosal adjuvant in whole inactivated virus (WIV)-based nasal vaccine. The results revealed that CS-IONzyme increased antigen adhesion to nasal mucosa by 30-fold compared to H1N1 WIV alone. Moreover, subunit vaccines may not elicit a robust CD8+ immune response against intracellular pathogens [97]. Interestingly, the nanozyme could also act as immunostimulatory molecules to trigger the defensive mechanism against SARS-CoV-2. Therefore, in order to overcome the above ongoing challenge for the development of an effective vaccine and ensure its availability all over the world, a nanozyme could play a significant role. Thus, in future vaccine development companies must benefit from the characteristics of nanozymes for SARS-CoV-2 vaccines.

## 4. Conclusions and Recommendations

Nanotechnology is expected to play a critical role in the fight against COVID-19. Various nanomaterials have been utilized for therapeutics, building rapid diagnostic test kits, inhibition of virus replication and vaccines. Morphological and physicochemical similarities of nanozymes with SARS-CoV-2 mean they can provide powerful tools to interfere with the viral life cycle. The multi-functionality of nanozymes will significantly promote the proficient treatment against SARS-CoV-2. Nanozymes can directly inhibit the entry of the virus into the host cells by blocking the attachment or inhibiting the viral replication. IONzymes have to an extraordinary extent improved the defensive capacity of PPEs like facemasks, specifically by halting the viral actions. Hence, efficient antiviral strategies are vital to minimize viral proliferation, cellular damages induced by viral invasions and mutation frequency, which otherwise may result in therapeutic resistance. Future research projects must explore the different combinations of biocompatible nanozymes to broaden the antiviral spectrum against SARS-CoV-2 and other human-infecting viruses. Moreover, efficacy of SARS-CoV-2 vaccines could be improved through the addition of immunomodulating nanozymes or using them as adjuvants. Nanozyme-based virus-like particles (VLPs) that mimic the SARS-CoV-2 can induce long-lasting immunity by avoiding exposure to virulent components. High thermostability and large-scale production of SARS-CoV-2 vaccines along with availability to all countries is the target area for future applications. In future, nanozymes can also be employed for lessening the impact of other global challenges like antimicrobial resistance [98,99]. Despite the incredible features of nanozymes, nanotoxicity and low selectivity are the limitations for biomedical applications [100]. Therefore, more deep research could improve the efficacy of antiviral medications and reduce their side effects. In summary, nanozyme/nanomaterial-based therapeutics are expected to play a frontline role in tackling this outbreak.

## Figures and Tables

**Figure 1 nanomaterials-11-01328-f001:**
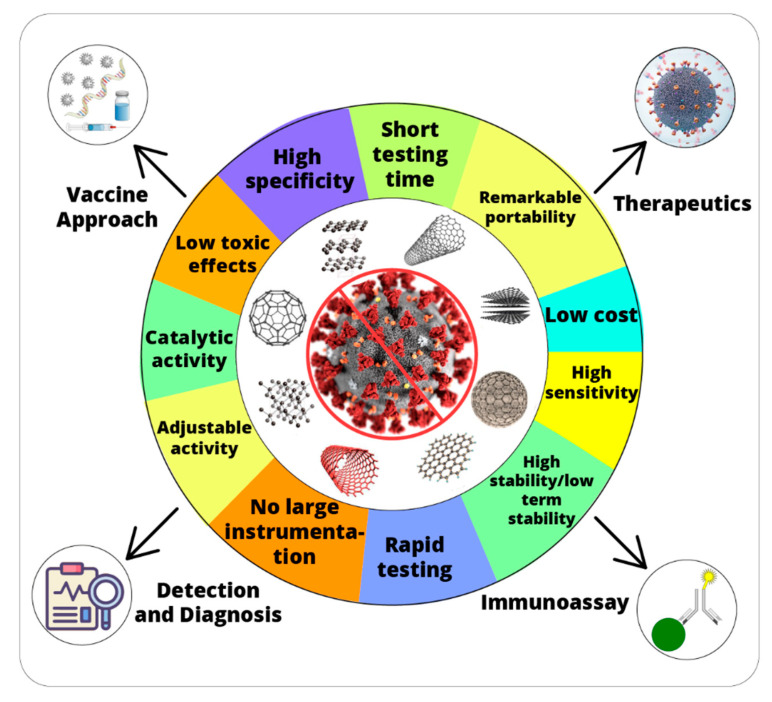
Advantages of nanozyme and their potential applications in fight against coronavirus disease 2019 (COVID-19).

**Figure 2 nanomaterials-11-01328-f002:**
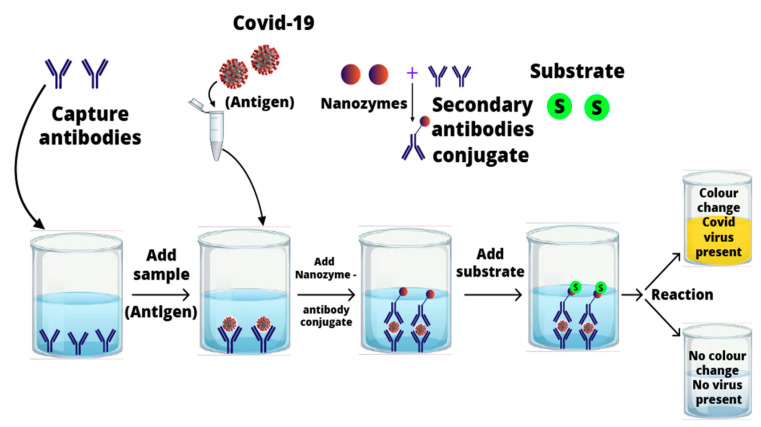
Possible mechanism of nanozyme-mediated sandwich enzyme-linked immunosorbent assay (ELISA) for SARS-CoV-2 detection.

**Figure 3 nanomaterials-11-01328-f003:**
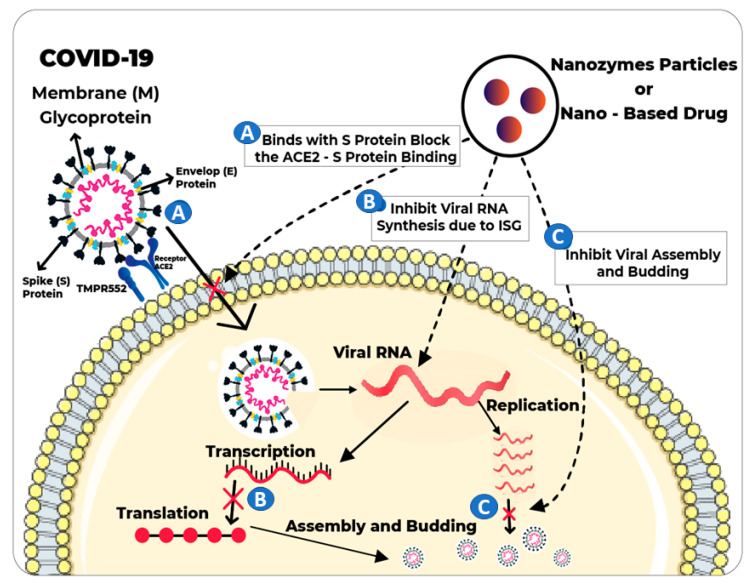
Possible mechanism of virus inactivation: (**A**) by blocking the viral entry, (**B**) by inhibiting viral RNA synthesis, (**C**) by blocking viral assembly and proliferation.

**Table 1 nanomaterials-11-01328-t001:** Different diagnostic techniques currently being used for severe acute respiratory syndrome coronavirus 2 (SARS-CoV-2).

Sr.No.	Technique	Sensitivity	Specificity	Limit of Detection	Cost in USD	Processing Time	Sample	Detection Element (Gene/Antibodies)	Reference
***Nucleic Acid-Based Techniques***									
1	RT-qPCR	80%	100%	0.0278 copies/µL	1.29–4.37	2–8 h; >12 h	Nasopharyngeal aspirate	N gene	[25,26,27]
2	Droplet Digital PCR (ddPCR)	83–99%	48–100%	5 × 10^3^ copies/µL	100	2–4 h	Throat swab samples	ORF1ab and N gene	[27,28]
3	RT-LAMP	97.50%	99.70%	1 × 10^2^ copies/µL	∼7	1 h	Pharyngeal swab	N gene	[29,30]
4	Recombinase Polymerase Amplification (RT-RPA)	65–94%	77–100%	0.05 copies/µL	4.3	15–20 min	Swab Samples	N gene, RdRp, E Gene	[31,32,33,34]
5	RCA-POC	99%	99%	1 copy/µL	4–10	<2 h	Nasopharyngeal swab	N gene and S gene	[35,36]
6	CRISPR	100%	100%	10 copies/µL	3.50	30–40 min	Oropharyngeal swab and Nasopharyngeal swab	E (envelope) and N (nucleoprotein) genes	[37,38]
7	RdRp/Hel RT-PCR	100%	100%	0.56 copies/µL	NA	<1 h	Respiratory and non-respiratory tract specimens	RdRp/Hel, spike (S), (N), genes	[39,40]
8	DETECTR (DNA Endonuclease-Targeted CRISPR Trans Reporter)	95%	100%	10 copies/μL	~0.2	40 min	Pharyngeal swab	E/N	[32,41]
9	Nucleic Acid Sequence-Based Amplification (NASBA)	89%	98%	0.5–5 copies/µL	3.66 and 12.61	10–50 min	Saliva	S gene	[42,43]
***Serological Based Techniques***									
10	ELISA	99.30%	95–96%	100 pg/ml	71.40	3–5 h	Blood samples	IgG, IgM	[27,44,45]
11	IFA (Immunofluorescence Assay)	64.50%	96.3–100%	NA	NA	3 h	Serum samples	IgG, IgM	[46,47]
12	Neutralization Assay	76.5–100%	100%	0.22 copies/µL	NA	2–3 days	Serum samples	IgG, IgM	[48,49]
13	Antigen Detection Assay	75.50%	94.90%	46–750 copies/µL	5	15 min	Nasopharyngeal	Nucleocapsid	[50,51,52]
14	Chemiluminescence Enzyme Immunoassays (CLIA)	99.67/90%	99.77/80%	NA	NA	45 min	Plasma	SARS-Cov2 RBD	[53,54]
15	WB (Westren Blotting)	90.90%	98.30%	NA	NA	4 h	Serum	Antinucleocapsid antibody	[27,55,56]
16	HTP-Microfluidic Device	95%	91%	1.6 ng/mL	NA	2.5 h	Serum samples	IgG, IgM	[57]
17	Lateral Flow Assay	96.7–100%	97.5–98.8%	5–20 ng/mL	27.42	15 min	Blood samples	IgG, IgM	[58,59,60]
18	Luciferase immunosorbent assay	96.7–100 %	100%	0.4–75 pg/μL	NA	<2.5 h	Serum samples	IgG	[61]
19	MIA (Microsphere Immunoassay)	90.17%	99.49%	0.121 U/L–0.366 U/L	NA	10–15 min	Serum samples	IgG, IgM, and IgA	[62,63]
20	Immunochromatographic Assay	43.20%	98.0%	0.18 ng/μL	NA	10–15 min	Serum samples	IgG, IgM	[64,65]

**Table 2 nanomaterials-11-01328-t002:** Nanozymes used in different antiviral therapies.

Sr. No.	Nanozymes	Disease/Pathogen	Application/Principle	References
**1.**	Metal nanozymes (Au-Pt core-shell nanozyme) Peroxidase-mimicking MNP_3_	Infectious diseases such as SARS-CoV, MERS-Covid-19.	Paper base biosensors for colorimetric detection.	[70]
**2.**	Co-Fe@hemin-peroxidase nanozyme	SARS-CoV-2	Chemiluminescence paper test for rapid and sensitive detection of SARS-CoV-2 antigen.	[71]
**3.**	Fe_3_O_4_ magnetic nanoparticle (Nanozyme probe)	Ebola virus	Detects the glycoprotein of Ebola virus.	[72]
**4.**	MagLISA	Influenza virus A	Provide detection, reduce spread of influenza virus and provide immediate clinical treatment.	[73]
**5.**	Au@Pt Mesopotous SiO_2_ nanozymes	Mumps virus	Diagnosis of mumps-virus, more sensitive compared to conventional immunoassay.	[74]
**6.**	Nanozyme Aptanser	Human nanovirus (NoV)	Allows ultrasensitive NoV detection rapidly, offering simplicity of use.	[75]
**7.**	Fe, O_4_ Bi_2_, S_3_ nanozyme	Cancer cell	Induce cancer cell death, Anti-cancerous.	[76]
**8.**	Iron Oxide nanozyme	Influenza virus	Catalytic inactivation of influenza virus, provide protection from viral transmission and infection.	[77]

## Data Availability

Not applicable we did not have conducted any research experimental work in this study and don’t have any data avaialable or archived datasets or supplementary data.

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
