# Peer review of "Unveiling the Potential Role of Nanozymes in Combating the COVID-19 Outbreak"

_nanomaterials, 2021, doi:10.3390/nano11051328_

Round 1

Reviewer 1 Report

Authors provided interesting perspective on Nanosymes as covid-19 diagnosis and treatment. I have following minor comments. 

  1. Are schematic/figures made by authors or are copied from other papers?
  2. Any product based on nanosymes in the market ? if so provide the information, 

Author Response

Response to Reviewer 1 Comments
Authors provided interesting perspective on Nanosymes as covid-19 diagnosis and treatment. I have following minor comments. 
Comment 1: Are schematic/figures made by authors or are copied from other papers?

Authors:  All the schematics/figures were made by the authors.

Comment 2: Any product based on nanosymes in the market ? if so provide the information, 
Authors: There are various market products and patents based on the nanozymes such as, Iron nano oxides are being used to give peroxidase activity. Moreover, Black flower nanozyme cleaner (Product Id # 229319) used as H2O2 alternative in cleaning applications. Royal-Grow Products' Enzyme Max Organic Biological used as biofertilizers.

Reviewer 2 Report

This Perspective is very interesting offering to the reader several stimula to the advancement of nanosystems  to fight against COVID-19. The paper is well organized and is easy to read also for non experts in the field. I appreciated the collection of data in tables.

References need to be carefully controlled and completed in some case for example reference 5. Some reference reports BioRxiv or med Rxiv, it is better to give the original journal where the paper was published.

More important, is to carefully control the literature in order to report all relevant papers as for example R. De Gasparo et al. Nature 2021, doi.org/10.1038/s41586-021-03461-y.

I also suggest to increase the size of the central part of Figure 1 so to make it more easy to read.

Figure 3. Transcrippion should be Transcription.

In conclusion, with minor changes I strongly support the publication of the manuscript.

Author Response

Response to Reviewer 2 Comments

This Perspective is very interesting offering to the reader several stimula to the advancement of nanosystems  to fight against COVID-19. The paper is well organized and is easy to read also for non-experts in the field. I appreciated the collection of data in tables.

Authors:  Thank you so much for appreciating and positive comments about our work.

Comment 1: References need to be carefully controlled and completed in some case for example reference 5. Some reference reports BioRxiv or med Rxiv, it is better to give the original journal where the paper was published.

Authors:  The references from published journal studies (where available) have been updated throughout the manuscript, as suggested by the reviewer. 

Comment 2: More important, is to carefully control the literature in order to report all relevant papers as for example R. De Gasparo et al. Nature 2021, doi.org/10.1038/s41586-021-03461-y.

Authors:  We have checked and have inserted all the relevant references including the above research article as per suggestions of the reviewer.

Comment 3: I also suggest to increase the size of the central part of Figure 1 so to make it more easy to read.

Authors:  Figure.1 was replaced with a new one that includes the modified font and text of the central part; please check the revised version of manuscript file.

Comment 4: Figure 3. Transcrippion should be Transcription.

Authors:  The term Transcrippion was replaced with Transcription in Figure.3

Comment 5: In conclusion, with minor changes I strongly support the publication of the manuscript.

Authors:  Thank you for your strong support of our manuscript. The conclusion section was revised and future prospects were added to update this section.

Reviewer 3 Report

The manuscript entitled "Unveiling potential role of Nanozymes in Combating the COVID-19 outbreak" by Jafar Ali and other is additional and routine manuscript of COVID management scenario however here authors highlighted the efficacy of nanozyme system in combating the covid.

1) why authors mixed the flow of information with general aspects to treatment options of covid specially in second page paragraph 2.

2). Hypothesis part in first paragraph of 3 page is completely mishandled, scientific highness is required. Coining of the wordings are just misleading and compete rewrite is required.

3). table one can be rearranged based on the serological, RNA and cell based assay

4). section 3.2 should be elaborately discussed with proper scientific references.

5). 3.2.1 what inhibit viral entry and blocks the biding. A wage statement.

6). Information from figure 3 is not adequately discussed in the text.

7). conclusions section cab ne modified with highlighting the proper future prospects.

Author Response

Response to Reviewer 3 Comments

The manuscript entitled "Unveiling potential role of Nanozymes in Combating the COVID-19 outbreak" by Jafar Ali and other is additional and routine manuscript of COVID management scenario however here authors highlighted the efficacy of nanozyme system in combating the covid.

Comment 1: why authors mixed the flow of information with general aspects to treatment options of covid specially in second page paragraph 2.

Authors:  The text was revised in the mentioned paragraphs in the introduction section to facilitate the flow of information as requested. Please check the revised version of the manuscript.

Comment 2: Hypothesis part in first paragraph of 3 page is completely mishandled, scientific highness is required. Coining of the wordings are just misleading and compete rewrite is required.

Authors:  Hypothesis section of the manuscript has been updated with scientific information as requested. Kindly check the last section of introduction at page # 3

Comment 3: table one can be rearranged based on the serological, RNA and cell based assay

Authors:  The table 1 has been rearranged in revised version of the article based on the nucleic acid based and serological assays. .

Comment 4: section 3.2 should be elaborately discussed with proper scientific references.

Authors:  The discussion in section 3.2 was elaborated and new references have been inserted as suggested by the reviewer.  

Comment 5: 3.2.1 what inhibit viral entry and blocks the biding. A wage statement.

Authors:  The heading/statement has been revised as ‘‘Inhibition of viral entry’’

Comment 6: Information from figure 3 is not adequately discussed in the text.

Authors:  The figure 3 depicts the use of Nanozymes for inactivation/ inhibition of SARS‐CoV‐2, and it has been discussed and referred frequently in section 3.2.1 and 3.2.2. However brief discussion was added to emphasize the importance of figure 3.

Comment 7: conclusions section cab ne modified with highlighting the proper future prospects.

Authors:  The conclusion section is revised and future prospects of the current article were also highlighted as suggested by the reviewer.
